# Do They Really Work? Quantifying Fabric Mask Effectiveness to Improve Public Health Messaging

**DOI:** 10.3390/ijerph19116372

**Published:** 2022-05-24

**Authors:** Charles Freeman, Reuben Burch, Lesley Strawderman, Catherine Black, David Saucier, Jaime Rickert, John Wilson, Holli Seitz, Jeffrey Stull

**Affiliations:** 1School of Human Sciences, Mississippi State University, Starkville, MS 39762, USA; cblack@humansci.msstate.edu; 2Department of Industrial & Systems Engineering, Mississippi State University, Starkville, MS 39762, USA; burch@ise.msstate.edu (R.B.); strawderman@ise.msstate.edu (L.S.); 3Human Factors & Athlete Engineering, Center for Advanced Vehicular Systems, Mississippi State University, Starkville, MS 39759, USA; dns105@cavs.msstate.edu; 4Institute for Clean Energy and Technology, Mississippi State University, Starkville, MS 39759, USA; rickert@icet.msstate.edu (J.R.); wilson@icet.msstate.edu (J.W.); 5Department of Communication, Mississippi State University, Starkville, MS 39762, USA; hseitz@comm.msstate.edu; 6International Personal Protection, Incorporated, Austin, TX 78709, USA; jeffstull@intlperpro.com

**Keywords:** filtration efficiency, barrier face covering, healthcare workers, COVID-19

## Abstract

The purpose of this study is to compare masks (non-medical/fabric, surgical, and N95 respirators) on filtration efficiency, differential pressure, and leakage with the goal of providing evidence to improve public health messaging. Masks were tested on an anthropometric face filtration mount, comparing both sealed and unsealed. Overall, surgical and N95 respirators provided significantly higher filtration efficiency (FE) and differential pressure (dP). Leakage comparisons are one of the most significant factors in mask efficiency. Higher weight and thicker fabric masks had significantly higher filtration efficiency. The findings of this study have important implications for communication and education regarding the use of masks to prevent the spread of COVID-19 and other respiratory illnesses, specifically the differences between sealed and unsealed masks. The type and fabric of facial masks and whether a mask is sealed or unsealed has a significant impact on the effectiveness of a mask. Findings related to differences between sealed and unsealed masks are of critical importance for health care workers. If a mask is not completely sealed around the edges of the wearer, FE for this personal protective equipment is misrepresented and may create a false sense of security. These results can inform efforts to educate health care workers and the public on the importance of proper mask fit.

## 1. Introduction

Throughout the COVID-19 global pandemic, one of the most contested public health guidelines is the recommendation for wearing a non-medical fabric mask by the public. Recently, the Delta variant of the COVID-19 virus has reignited precautionary measures for the vaccinated and unvaccinated. With many states and agencies reimplementing mask mandates, the public outcry surrounding their effectiveness is fraught with distrust and misinformation. The Centers for Disease Control and Prevention (CDC) and the World Health Organization (WHO) have been responsible for managing shifting guidelines [1,2] as more knowledge from research regarding the pandemic is shared. The shift in messaging is understandably part of the reasoning behind why these guidelines have seen resistive adoption [3,4,5]. Additionally, a lack of standardized test procedures for filtration efficiency of fabric masks or minimum performance requirements has severely impacted public messaging. In February 2021, in coordination with the CDC, ASTM International released *F3502—21 Standard Specification for Barrier Face Coverings* [6] as the first standard to address non-medical fabric masks in response to the global pandemic. The intent of the standard is to provide testing standardization repeatability to improve communication and clarity for the public with respect to the filtration efficiency of barrier face coverings, more commonly known as non-medical fabric masks [6]. In this study, the ASTM International testing procedure is applied using a face filtration mount to evaluate non-medical fabric masks, surgical masks, and N95 respirators in comparison with the newly established benchmarks. A unique assessment addition of critical importance for evaluating non-medical fabric mask efficacy is the evaluation of leakage impacts for all types of face coverings and the impacts on messaging to the public.

Barrier face coverings are defined as a method of source control to reduce the number of aerosol droplets exhaled by an individual wearer and to provide some level of particulate filtration for aerosol droplets inhaled [1,6]. Prior to the recent filtration efficiency testing standard (ASTM F3502), health organizations released guidance for health service professionals and the public on wearing face coverings. Generally, health organizations advised all persons who are not able to physically distance to wear a barrier face covering and those in medical care environments to wear certified N95 respirators or surgical masks [2,6,7]. However, in many of these instances, proper fitting and leakage minimization is not achieved, leaving filtration efficiencies (FE) lower than the minimum requirement of 90% FE [8,9]. While recent studies have examined the aerosol penetration of both flat and structured materials [10,11,12,13,14,15,16,17], variations across methods contributed to the need for standardized testing. A challenge for comprehensive and consistent testing is the inability to effectively measure a non-structured fabric mask, as opposed to the more rigid N95 respirator or flat surgical mask. Therefore, prior studies’ results are variable and may be misleading to the overall public understanding of mask filtration and effectiveness. A recent study following ASTM F3502 evaluated filtration of commercial fabric masks. Researchers developed a face mount apparatus based on the National Institute of Occupational Safety and Health (NIOSH) anthropometric head forms to test FE. When the masks were only sealed at the facial contact points, none of the masks in the study reported above the 20% FE minimum threshold [14]. However, ASTM F3502 requires masks to be completely sealed around the edges to prevent leakage and more accurately measure the true FE for both inflow and outflow [6]. In simplistic terms, the reality of mask wearers to properly seal all sides and edges of the mask completely to their face seems unlikely. Therefore, to use a standard that tests masks completely sealed to a fixture is not in alignment with everyday application and minimizes the negative impact leakage can have on mask efficacy [14].

To date, the research remains limited on standardized testing method outlined by ASTM International to evaluate (a) FE, (b) differential pressure (dP), and (c) leakage to compare non-medical masks with N95 respirators and surgical masks. However, recent publications have utilized more rigorous methods for evaluation of non-medical-grade masks [18,19]. In the aftermath of shifting public health recommendations on mask wearing, the lack of established research protocols for results dissemination significantly contributes to adoption variance [14,15,20]. While many states and countries are lifting mask mandates, the CDC projects mask-wearing through 2022, especially during flu seasons. Without consistency in mask-testing methods, the effectiveness of different mask coverings in mitigating the spread of COVID-19 cannot be effectively communicated to the public. Confusion about the effectiveness of masks, exacerbated by inconsistent recommendations early in the pandemic [21], may be associated with lack of adherence to masking recommendations, and inconsistencies in messaging about face masks foster mistrust of public health authorities [22]. The public has expressed a desire for scientific evidence on the effectiveness of masks (particularly cloth face coverings) to assist in decision-making [22]. Therefore, the purpose of this study is to evaluate face coverings (non-medical/fabric, surgical, and N95 respirators) on FE, dP, and leakage with the goal of providing evidence to improve public health messaging. A major outcome of this study is to provide additional recommendations of communication in conjunction with the ASTM F3502 standards for proper quantification of leakage impacts when all masks are tested on an anthropometric head form in a Model 8118A salt aerosol generator.

## 2. Materials and Methods

### 2.1. Face Barrier Selection

Per the recommendations provided by the CDC [1] and WHO [2] and prior face barrier filtration studies [7,8,9,10,11,12,13,14,16,17], researchers selected 13 face barriers (identified as F1–F13), all containing multiple layers with the surgical and N95 respirators certified by a third party and authorized for use in COVID-19 medical response environments. All the 11 nonmedical face barriers contained two or more layers, with F4 and F11 containing a polyester non-woven material for filtration. F1 and F6 are the only masks not commercially available and were constructed using a sewing pattern drafted from the head forms to maximize fit. These two would represent the homemade face barrier options available. From the face barriers tested, seven, including the surgical and N95, contain 100% synthetic fibers (polyester, polypropylene, nylon, and spandex); four contain a varying blend of natural and synthetic fibers across layers (cotton, viscose, polyester, nylon, and spandex); and two contain 100% natural fibers for all layers (cotton). Materials construction across all layers for nine of the face barriers tested are knit, knit/woven, or knit/non-woven combinations; two are woven, and two are non-woven (surgical and N95). See Table 1 for a complete breakdown of face barrier characteristics.

### 2.2. Face Barrier Characteristics

Fabric construction and thread count/gauge characteristics used ASTM D3775-17e1 Standard Test Method for End (Warp) and Pick (Filling) Count of Woven Fabrics [23] and ASTM D8007-15 (2019) Standard Test Method for Wale and Course Count of Weft Knitted Fabrics [24]. Researchers calculated material weight in grams per square meter (GM2) using a 100 cm^2^ sample cutter and followed procedures outlined in ASTM 3776 Standard Test Methods for Mass Per Unit Area (Weight) of Fabric [25]. GM2 was calculated for outer, inter, and lining layers (where applicable) and combined for a total GM2 rating for each of the face barriers to be used for statistical analysis. Researchers followed ASTM D1777—96 Standard Test Method for Thickness of Textile Materials [26] to determine face barrier (all layers) thickness with a calibrated digital compression apparatus. See Table 1 for a complete breakdown of face barrier characteristics.

### 2.3. Face Mount Apparatus

ASTM F3502 [6] requires a full seal around the edges of face barriers tested laying flat against the filter holder or a mesh screen, wire frame, or similar device to prevent the collapse of the material into the filter holder. Testing recommendations indicate the use of support materials to mimic the facial structure of the nose and mouth. Any structure used should have at least 70% open area to support specimens, without adversely affecting the results. Fitting and sizing guidelines indicate support for manufacturers to utilize the NIOSH anthropometric data as a resource for product size development. Based on procedural limitations for the current test method, researchers designed a face mount apparatus based on the NIOSH anthropometric data for standard head forms and sizes. The medium-size head form was used and adapted to include a nasal passageway and open mouth with a hollowed interior to minimize the effect of results of face barrier evaluation Z (see Figure 1). Preliminary results without a face barrier indicate that the face mount apparatus effectively tests the filtration efficiency and differential pressure using the instrument and procedures outlined in 42 CFR Part 84 Standard Procedures [20] on flow rates and conditions for testing and certifying air-purifying and particulate respirators. Addition of the face mount apparatus does not adversely affect the results for filtration or differential pressure. Results generated from aerosol production followed specifications and standards outlined a Model 8118A salt aerosol generator (NaCl). Figure 1 shows the face mount apparatus without a mesh addition and face barrier fittings that was used for the testing procedures. Both face mounts (Figure 1 and Figure 2) are the same model and 3D print filament, despite the difference in colors. The mesh stabilization feature was added per the testing procedures in ASTM F3502.

### 2.4. Filtration Testing Methods

Specimens were preconditioned in accordance with ASTM F3502 Section 8.1.1.5 [6]. Following preconditioning, specimen mounting and setup followed two separate methods. Initially, specimens were mounted to the face mount apparatus and sealed, using double-sided men’s grooming tape (Vapon® Topstick®; Fairfield, NJ, USA), only at the major points of facial contact: nose, chin, and jawline (see Figure 2). These represent the contact points of a typical mask wearer based on prior mask fit research [27]. For the second round of testing, all specimens were completely sealed, using double-sided men’s grooming tape (Vapon® Topstick®; Fairfield, NJ, USA), and secured to the face mount apparatus. This follows the procedures for filtration testing outlined by ASTM International despite these conditions rarely being met in everyday use. In both conditions, specimens were tested for inflow and outflow filtration efficiency and differential pressure. The face filtration mount was sealed to the filtration adapter plate using a hot melt glue. A cylindrical chamber was created around the device and sealed to the adapter plates. Face barrier testing was performed using the TSI 8130A with the chamber and face filtration mount. Testing procedures used ASTM F3502 [6] and 42 CFR Part 84 Standard Procedures [20] to set the flow rates and conditions for testing and certifying air-purifying and particulate respirators. Aerosol production followed specifications and standards outlined with a Model 8118A salt aerosol generator. Sodium aerosol (NaCl) specifications included particles with a mass mean particle diameter of 0.26 µm and a count median particle diameter of 0.075 µm. Flow rates for both particle sizes followed standardized rates at ~85.0 L/min, much higher than the normal breathing rate of 40.0–60.0 L/min. Baseline readings were taken with the face filtration mount and no facemask installed. Results from baseline testing indicate minimal particle obstruction or distortion in the testing chamber for NaCl (0.08% FE, 0.0 dP). Means across five repeated data collection events for each of the mask types for FE and dP are used for statistical analysis as opposed to the lowest rates reported rounded to the nearest integer, because the scope of this project is to not to certify any brand/type of face barrier. Five masks for each type of mask ID were sourced or made and tested for a total of 65 specimens (5ea × 13 Mask IDs).

### 2.5. Data Analysis and Statistics

Due to non-normality of the data, Kruskal–Wallis tests were conducted to determine if there were differences in outcomes across fabric types. For all tests, an alpha value of 0.05 was used. Post hoc analysis was conducted through a series of pairwise comparisons, with Bonferroni adjustment for multiple comparisons. Subgroup designations for thickness and GSM are based on ASTM International recommendations and common practices in textile mechanical performance research.

## 3. Results

For all the figural and statistical representations, researchers tested 65 total masks (5 specimens × 13 mask types) over four testing conditions (sealed inflow, sealed outflow, unsealed inflow, and unsealed outflow) for a total of 260 filtration and differential pressure tests.

### 3.1. Overall Mask Comparison—Sealed Mask Setting

Filtration efficiency was significantly different based on fabric type for both the outflow (χ^2^(11, N = 58) = 52.718, *p* < 0.001) and inflow (χ^2^(11, N = 58) = 53.465, *p* < 0.001) in experimental settings. Overall, the surgical mask and N95 mask had significantly higher filtration efficiency than all fabric masks tested. There was no significant difference found between surgical mask and N95 mask regarding filtration efficiency. With respect to the fabric masks, filtration efficiency was highest for fabrics 1 and 11, and lowest for fabrics 3, 4, 5, and 10. Other fabric types did not demonstrate a significant difference in post hoc analyses.

Differential pressure was significantly different based on fabric type for both the outflow (χ^2^(11, N = 58) = 53.399, *p* < 0.001) and inflow (χ^2^(11, N = 58) = 52.559, *p* < 0.001) in experimental settings. Overall, differential pressure was highest for fabric masks 1, 6, and 9, and lowest for fabric masks 3 and 4. Again, the surgical and N95 masks were higher than all of the fabric masks. Other fabric types did not demonstrate a significant difference in post hoc analyses.

### 3.2. Sealed vs. Unsealed

Mask performance for the sealed condition was compared to the unsealed condition for the outflow condition using NaCl aerosol. Both filtration efficiency and differential pressure were significantly different based on whether the mask was sealed or unsealed on the experimental apparatus. Filtration efficiency (Table 2) was significantly higher in the sealed condition for fabric masks 1, 2, 3, 4, 6, 8, 9, and 11, as well as the surgical and N95 masks. Figure 3a,b visualize the significant differences between sealed and unsealed for fabric masks and surgical and N95 masks.

Differential pressure (Table 3) was significantly higher in the sealed condition for fabric masks 1, 2, 5, 6, 8, 9, 10, and 11, as well as the surgical and N95 masks, with the majority of differences realizing a large effect size. The differential pressure of fabric 3 was significantly higher in the unsealed experimental condition. See Figure 3a for comparisons of sealed versus unsealed for the fabric masks and Figure 3b for comparisons between N95 and surgical masks.

### 3.3. Outflow vs. Inflow

Mask performance based on flow condition was compared for the sealed condition using NaCl aerosol. Only three mask types demonstrated a significant difference in filtration efficiency based on air flow direction (Table 4): fabric mask 3, surgical mask, and N95 mask. Flow direction impacts under sealed conditions are critical to analyze based on significant leakage occurring during outflow of unsealed masks. Inhalation of virus-laden particles is typically less likely when wearing a mask; however, significant viral particles are expulsed during unsealed exhalation. All other fabric types showed no significant difference in filtration efficiency based on airflow direction. Differential pressure (Table 5) was significantly higher in the inflow condition for all fabric types, with the majority of differences realizing a large effect size.

### 3.4. Comparison by Fabric Mask Characteristics

Mask performance based on fabric mask type was compared for the sealed condition using NaCl aerosol. The masks were categorized based on weight (GSM—grams per square meter), thickness (mm), and fiber (natural, synthetic, or blend). Results are shown in Table 6 (outflow condition) and Table 7 (inflow condition). Regarding fabric weight, fabrics with a higher weight had significantly higher filtration efficiency for both airflow conditions. Fabric thickness presented similar results, with significantly higher thickness fabrics having significantly higher filtration efficiency measurements for both airflow conditions. There was no significant difference of fabric weight or thickness on differential pressure.

Regarding the type of fiber used in the mask fabric, natural fiber masks had significantly lower filtration efficiency than synthetic or blend masks. This was found for both airflow directions. When evaluating differential pressure, synthetic masks had significantly lower measurements than natural or blend masks. This was also found for both airflow directions.

## 4. Discussion

The ASTM F3502 standard test method to evaluate FE, dP, and leakage is outlined in this study and remains one of the earliest to follow the standard in the analysis of multiple types of masks. The study additionally provides a comparison to non-standard mask filtration testing. Results from the testing for fabric masks, surgical masks, and N95 respirators are compared in terms of aerosol particle penetration (0.3 µm) with a NaCl solution at 85 L/min. For non-standardized testing, masks were fitted to the face mount apparatus (see Figure 1) and adhered at contact points at the bridge of the nose, under the chin, and along the jawline for unsealed aerosol testing. In accordance with ASTM F3502, all masks were completely sealed to the face mount apparatus for standardized testing purposes around the edges using a double-sided adhesive. Masks were challenged with aerosol particles with the face mount facing up to evaluate the wearer filtration and exposure from external sources. The face filtration mount was inverted, and masks were challenged to mimic the wearer exhalation filtration. Both inhalation and exhalation testing were performed for both unsealed and sealed masks. Table 2 validates the testing method procedure and use of the face filtration mount due to the sealed surgical mask and N95 respirators reporting FE above 90%, which is in accordance with the testing procedure and standards previously provided by NIOSH. Of the remaining masks tested, only F1 and F11 met the minimum standard >20% FE established by CDC and ASTM International. When unsealed, only the N95 met this requirement. All masks tested met the minimum <15 mm H_2_O dP, indicating no limitations with breathability on the masks tested (see Table 3). Except for F5 and F10, all masks reported a significant difference between the unsealed and sealed FE, with the surgical and N95 reporting two of the largest effect sizes. Table 5 indicates limited significant FE differences between inflow and outflow testing. Therefore, the FE data for masks are similar for both inhalation and exhalation, further supporting the compounding effect of both parties wearing a mask. However, there are significant differences in dP between inflow and outflow testing, which is supported by the design of the face mount apparatus limiting exhalation outflow. Overall, this study confirms the validity of ASTM F3502 for evaluating sealed masks yet may not provide a real-world analysis of mask usage. This study identifies that the crucial area of protection is in the reduction of leakage effects and supports proper secured fitting to the wearer. Although the rollout of certification of testing for masks is a step in the right direction for increasing public acceptance of usage, this study shows that there are significant limitations for mask manufacturers and wearers to meet these new standards.

## 5. Conclusions

These results have important implications for communication and education efforts related to the use of masks to prevent the spread of COVID-19 and other respiratory illnesses. In particular, findings related to differences between sealed and unsealed masks are of critical importance for health care workers. Specimens in this study are not comprehensive of all masks available on the market. However, results, while not widely generalizable, indicate the need for multiple levels of controls and safety to prevent the continued spread of respiratory infections. If a mask is not completely sealed around the edges of the wearer, FE for this personal protective equipment is misrepresented and may create a false sense of security. These results can inform efforts to educate health care workers and the public on the importance of proper mask fit. Further, data on the comparative effectiveness of masks can be used to fulfill the previously established public need for evidence-based guidelines and inform future public health guidance and public communication efforts. Finally, to increase transparency and clarity, messaging on mask effectiveness for the general public needs to be crafted in line with the performance requirements outlined in ASTM F3502. Consistent use of these guidelines in communicating mask effectiveness may have implications for increased public trust of and support for public health guidelines.

## Figures and Tables

**Figure 1 ijerph-19-06372-f001:**
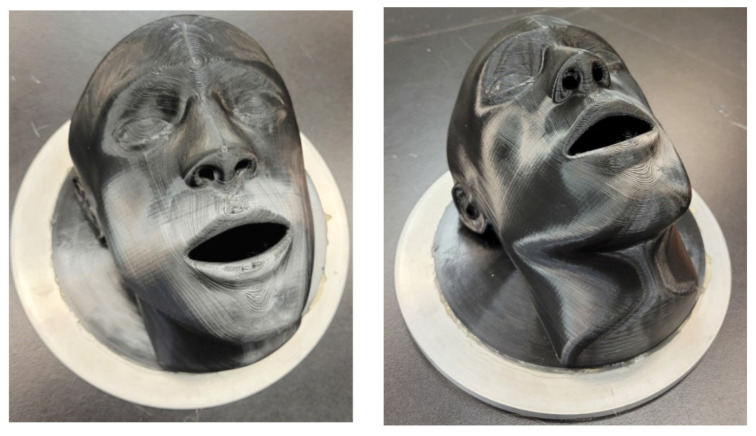
Face mount apparatus.

**Figure 2 ijerph-19-06372-f002:**
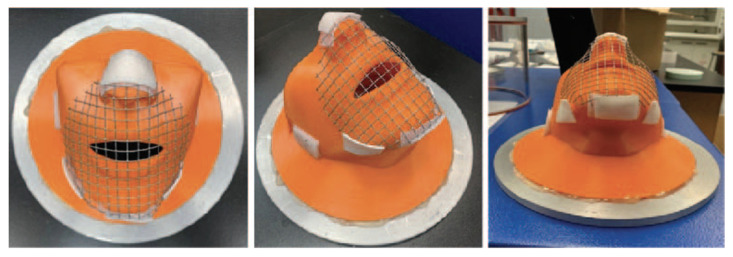
Face filtration mount with mesh cage.

**Figure 3 ijerph-19-06372-f003:**
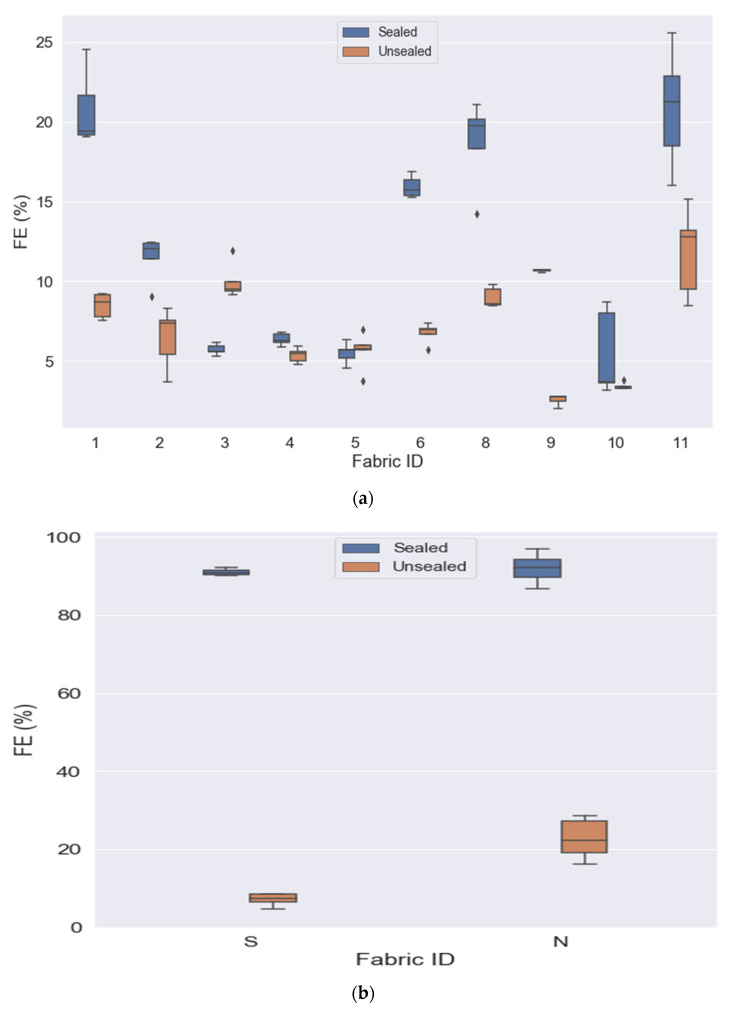
(**a**) Comparison of filtration efficiency of sealed vs. unsealed by fabric mask ID. (**b**) Comparison of filtration efficiency of sealed vs. unsealed for surgical (S) and N95 respirators (N).

**Table 1 ijerph-19-06372-t001:** Characteristics of masks used in the study.

Mask ID	Available to Purchase	Mask Layer	Fabric Structure	Fiber Content	Thread/ Loop (Per 10 cm^2^)	Mass (g/m^2^)	Thickness (mm)
F1	N ^1^	Outer	Tricot	82% nylon 18% spandex	293	296	1.05
F1	-	Lining	Single knit	87% cotton 13% polyester	153	-	-
F2	Y ^1^	Outer	Double knit	93% polyester 7% spandex	135	431	1.50
F2	-	Lining	Raschel warp knit	92% polyester 8% spandex	217	-	-
F3	Y	Outer	Double weft knit	100% polyester	195	278	0.97
F3	-	Lining	Single knit	50% viscose 50% cotton	170	-	-
F4	Y	Outer	Raschel warp knit	100% polyester	170	281	1.21
F4	-	Inter-lining	Non-woven	100% polyester	N/A	-	-
F4	-	Lining	Raschel warp knit	100% polyester	209	-	-
F5	Y	Outer	Plain Weave	100% cotton	310	277	0.62
F5	-	Lining	Single Knit	96% polyester 4% spandex	192	-	
F6	N	Outer	Single Knit	89% cotton 11% spandex	229	322	1.25
F6	-	Lining	Single knit	87% cotton 13% polyester	153	-	
F7	Y	Outer	Single knit	83% polyester 17% spandex	195	192	0.46
F8	Y	Outer	Double weft knit	100% polyester	212	320	0.99
F8	-	Lining	Double weft knit	100% polyester	211	-	-
F9	Y	Outer	Plain weave	100% cotton	245	241	0.61
F9	-	Lining	Plain weave	100% cotton	276	-	-
F10	Y	Outer	Plain weave	100% cotton	324	206	0.40
F10	-	Lining	Plain weave	100% cotton	324	-	-
F11	Y	Outer	Double knit	100% polyester	144	439	2.01
F11	-	Inter-lining	Open cell foam	100% polyurethane	N/A	-	-
F11	-	Lining	Double weft knit	77% nylon/23% spandex	234	-	-
Surgical	Y	N/A	Spun bond non-woven	100% polypropylene	N/A	25	0.4
N95	Y	N/A	Spun bond/ melt-blown non-woven	100% polypropylene	N/A	75	2.14

1 N, no; Y, yes.

**Table 2 ijerph-19-06372-t002:** Comparison of FE (%) between sealed and unsealed conditions.

Mask ID	FE (%) (M(SD))		Comparison	Effect Size (Cohen’s D)
	Sealed	Unsealed		
F1	20.840 (1.738)	8.482 (0.769)	*t*(4) = 13.32, *p* < 0.001	5.917
F2	11.592 (1.880)	6.486 (1.886)	*t*(4) = 8.552, *p* = 0.001	3.825
F3	6.931 (0.528)	9.987 (1.114)	*t*(4) = −5.111, *p* = 0.007	−2.286
F4	7.174 (0.486)	5.359 (0.458)	*t*(4) = 29.283, *p* < 0.001	13.096
F5	6.186 (1.339)	5.635 (1.168)	*t*(4) = 0.494, *p* = 0.647	0.221
F6	19.962 (5.532)	6.757 (0.628)	*t*(4) = 4.851, *p* = 0.008	2.170
F8	18.230 (2.758)	8.992 (0.621)	*t*(4) = 8.543, *p* = 0.001	2.418
F9	10.874 (1.284)	2.568 (0.332)	*t*(2) = 8.796, *p* = 0.013	5.079
F10	5.725 (3.120)	3.429 (0.237)	*t*(4) = 1.561, *p* = 0.194	3.290
F11	21.967 (2.997)	11.808 (2.755)	*t*(4) = 4.713, *p* = 0.009	4.820
S	91.304 (0.788)	7.064 (1.617)	*t*(4) = 97.157, *p* = <0.001	43.450
N95	90.427 (2.812)	22.598 (5.229)	*t*(4) = 20.871, *p* = <0.001	9.334

Note: M = mean; SD = standard deviation.

**Table 3 ijerph-19-06372-t003:** Comparison of dP (mm H_2_O) between sealed and unsealed conditions.

Mask ID	dP mm H_2_O (M(SD))		Comparison	Effect Size (Cohen’s D)
	Sealed	Unsealed		
F1	6.760 (0.462)	3.040 (0.428)	*t*(4) = 10.11, *p* = 0.001	4.521
F2	3.280 (0.335)	2.140 (0.114)	*t*(4) = 7.421, *p* = 0.002	3.319
F3	1.500 (0.122)	2.360 (0.445)	*t*(4) = −3.984, *p* = 0.016	−1.782
F4	1.640 (0.230)	1.340 (0.207)	*t*(4) = 1.604, *p* = 0.184	0.717
F5	2.920 (0.286)	1.840 (0.089)	*t*(4) = 8.703, *p* = 0.001	3.892
F6	5.680 (0.427)	1.640 (0.114)	*t*(4) = 21.719, *p* < 0.001	9.713
F8	4.100 (0.548)	1.880 (0.228)	*t*(4) = 7.929, *p* = 0.001	0.626
F9	11.633 (0.551)	1.300 (0.332)	*t*(2) = 39.307, *p* = 0.001	22.694
F10	5.860 (2.900)	1.180 (0.259)	*t*(4) = 3.749, *p* = 0.02	2.791
F11	2.940 (0.537)	1.360 (0.456)	*t*(4) = 4.427, *p* = 0.011	0.798
S	11.340 (0.888)	0.760 (0.182)	*t*(4) = 28.136, *p* = <0.001	12.583
N95	12.480 (1.602)	3.460 (0.760)	*t*(4) = 10.303, *p* = 0.001	4.608

Note: M = mean; SD = standard deviation.

**Table 4 ijerph-19-06372-t004:** Comparison of FE % based on air flow direction.

Mask ID	FE (%) (M(SD))		Comparison	Effect Size (Cohen’s D)
	Outflow	Inflow		
F1	20.840 (1.738)	20.766 (2.348)	*t*(4) = 0.091, *p* = 0.932	0.041
F2	11.592 (1.880)	11.483 (1.412)	*t*(4) = 0.104, *p* = 0.992	0.046
F3	6.931 (0.528)	5.717 (0.332)	*t*(4) = 6.388, *p* = 0.003	2.857
F4	7.174 (0.486)	6.362 (0.383)	*t*(4) = 2.419, *p* = 0.073	1.082
F5	6.186 (1.339)	5.500 (0.654)	*t*(4) = 1.881, *p* = 0.133	0.841
F6	19.962 (5.532)	15.930 (0.698)	*t*(4) = 1.623, *p* = 0.180	0.726
F8	18.230 (2.758)	18.709 (2.686)	*t*(4) = −1.62, *p* = 0.180	0.661
F9	10.874 (1.284)	10.675 (0.118)	*t*(2) = 0.27, *p* = 0.812	1.272
F10	5.725 (3.120)	5.443 (2.666)	*t*(4) = 0.578, *p* = 0.594	1.093
F11	21.967 (2.997)	20.822 (3.713)	*t*(4) = 1.219, *p* = 0.290	2.100
S	91.304 (0.788)	90.521 (0.404)	*t*(4) = 3.205, *p* = 0.033	1.433
N95	90.427 (2.812)	93.558 (2.874)	*t*(4) = −6.933, *p* = 0.002	−3.100

Note: M = mean; SD = standard deviation.

**Table 5 ijerph-19-06372-t005:** Comparison of dP (mm H_2_O) based on air flow direction.

Mask ID	dP mm H_2_O (M(SD))		Comparison	Effect Size (Cohen’s D)
	Outflow	Inflow		
F1	6.760 (0.462)	15.220 (1.994)	*t*(4) = −10.595, *p* < 0.001	−4.738
F2	3.280 (0.335)	5.820 (0.773)	*t*(4) = −10.772, *p* < 0.001	−4.817
F3	1.500 (0.122)	2.580 (0.084)	*t*(4) = −28.864, *p* < 0.001	−12.908
F4	1.640 (0.230)	2.840 (0.297)	*t*(4) = −12.649, *p* < 0.001	−5.657
F5	2.920 (0.286)	4.820 (0.536)	*t*(4) = −11.355, *p* < 0.001	−5.078
F6	5.680 (0.427)	8.840 (0.702)	*t*(4) = −18.685, *p* < 0.001	−8.356
F8	4.100 (0.548)	6.860 (0.623)	*t*(4) = −13.069, *p* < 0.001	0.472
F9	11.633 (0.551)	15.267 (0.379)	*t*(2) = −12.503, *p* = 0.006	0.503
F10	5.860 (2.900)	7.800 (3.621)	*t*(4) = −3.29, *p* = 0.030	1.318
F11	2.940 (0.537)	4.000 (0.704)	*t*(4) = −10.296, *p* = 0.001	0.230
S	11.340 (0.888)	13.800 (1.005)	*t*(4) = −20.359, *p* = <0.001	−9.105
N95	12.480 (1.602)	14.880 (2.153)	*t*(4) = −8.09, *p* = 0.001	−3.618

Note: M = mean; SD = standard deviation.

**Table 6 ijerph-19-06372-t006:** Comparison of results based on fabric mask type for outflow condition.

Mask Type		FE % (M(SD))	Comparison	dP mm H_2_O (M(SD))	Comparison
GSM	<300	9.532 (5.768)	*U*(48) = 481, *p* < 0.001	4.582 (3.450)	*U*(48) = 307, *p* = 0.572
	>300	17.938 (5.163)		4.000 (1.168)	
Thickness	<1.0 mm	9.477 (5.338)	*U*(48) = 458, *p* < 0.001	4.643 (3.408)	*U*(48) = 288, *p* = 0.992
	>1.0 mm	16.307 (6.594)		4.060 (1.953)	
Fiber	Natural	7.091 (2.969)	χ^2^(2, N = 48) = 16.676, *p* < 0.001	6.062 (3.843)	χ^2^(2, N = 48) = 18.800, *p* < 0.001
	Synthetic	13.576 (7.088)		2.545 (1.148)	
	Blend	17.464 (5.407)		5.240 (1.553)	

Note: M = mean; SD = standard deviation.

**Table 7 ijerph-19-06372-t007:** Comparison of results based on fabric mask type for inflow condition.

Mask Type		FE % (M(SD))	Comparison	dP mm H_2_O (M(SD))	Comparison
GSM	<300	8.963 (5.985)	*U*(48) = 478, *p* < 0.001	7.575 (5.494)	*U*(48) = 316, *p* = 0.451
	>300	16.736 (4.218)		6.380 (1.908)	
Thickness	<1.0 mm	9.081 (5.711)	*U*(48) = 455, *p* < 0.001	6.787 (4.179)	*U*(48) = 306.5, *p* = 0.695
	>1.0 mm	15.073 (5.995)		7.344 (4.623)	
Fiber	Natural	6.672 (2.779)	χ^2^(2, N = 48) = 16.132, *p* < 0.001	8.377 (4.666)	χ^2^(2, N = 48) = 21.205, *p* < 0.001
	Synthetic	12.903 (7.396)		4.070 (1.799)	
	Blend	16.060 (4.205)		9.960 (4.231)	

Note: M = mean; SD = standard deviation.

## Data Availability

All data are available in the main text.

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
