# Peer review of "Do They Really Work? Quantifying Fabric Mask Effectiveness to Improve Public Health Messaging"

_ijerph, 2022, doi:10.3390/ijerph19116372_

Round 1

Reviewer 1 Report

TO THE AUTHORS

This is an interesting paper about the implementation of a variant procedure for testing filtration efficiency and differential pressure of face masks and barrier face coverings, in order to evidence the importance of face fit for the overall performance of the protective equipment.

The presented method can find useful and practical application in the field of mask testing. The manuscript is informative and conclusions fit the reported results.

However, there are some improvements that authors should perform to improve clarity and reproducibility of the presented methods that are currently lacking clarity and essential information. Part of this information was supposed to be in a supplementary material which actually was not available for review. The structured analytical and statistical approach should be better detailed in a dedicated paragraph under the Mat&Methods section. The authors are also suggested to check the recent literature on the specific topic of mask filtration testing and assessment of air leaks at the face-mask interface, comparing their findings with other authors results on very similar approaches (e.g.: https://doi.org/10.1080/02786826.2021.1933377 and https://doi.org/10.3390/ijerph19063548)

Quality and clarity of figures including graphs should be improved.

A number of specific comments are provided in line with the text in the attached pdf file.

Author Response

Please see attached for responses to reviewer 1 on the detailed PDF.

Round 2

Reviewer 1 Report

Authors sufficiently addressed all comments from this reviewer. Manuscript clarity and methods reproducibility has increased. Despite three additional references were added to the text, bibliographic list was not updated and should be checked.

Author Response

The authors have added those 3 references.